# Social Customer Relationship Management and Organizational Characteristics

**Kateřina Kantorová [1] and Pavel Bachmann [2,\*]**

[1] Institute of Business Economics and Management, University of Pardubice,
53210 Pardubice, Czech Republic; katerina.kantorova@upce.cz

[2] Department of Management, University of Hradec Kralove, 50003 Hradec Kralove, Czech Republic

\* Correspondence: pavel.bachmann@uhk.cz; Tel.: +42-49-333-2378

**Abstract:** Social customer relationship management (SCRM) is a new philosophy influencing the relationship between customer and organization where the customer gets the opportunity to control the relationship through social media. This paper aims to identify (a) the current level of SCRM and (b) the influence of basic organizational characteristics on the SCRM level. The data were gathered through a questionnaire distributed to 362 organizations headquartered in the Czech Republic. The questionnaire comprised 54 questions focusing on the significance of marketing and CRM practices, establishing a relationship with the customer, online communities, the use of social media in marketing, and acquiring and managing information. Scalable questions with a typical five-level Likert scale were applied in the questionnaire. The results show that larger firms more often set up their own online communities and manage them strategically; moreover, they are able to manage information better. Contrariwise, small-sized organizations use social networks as a way to establish communication with the customer more than large-sized entities. The use of social media for marketing purposes is significantly higher in organizations oriented to consumer markets than in those oriented to business markets.

**Keywords:** customer relationship management (CRM); social media; social CRM; customer information, small and medium enterprises (SMEs)

---

## 1. Introduction

Today's relationship with the customer is increasingly influenced by a customer community living on social networks. The way the relationship with the customer is built has changed over the course of history. In this new environment, the customer can control their relationships with businesses and attain power to influence others in their social network [1–3]. Moreover, in the understanding of customers´ actual needs, social media are becoming a new phenomenon. Standage [4] outlines social media as an environment in which information is "passed from one person to another along social connections, to create distributed discussion or community." "The blogs are new pamphlets, microblogs and online social networks are the new coffee houses and media sharing sites are the new commonplace books." Capturing data with such social media is—with the use of Facebook, YouTube, LinkedIn, Twitter, or/and blogs—easier than it ever was before. The new term "social customer relationship management" (SCRM) is used to distinguish between the use of all data, including "social" data, and the older approach based mainly on point-of-sale data [5–7]. Van Looy [8] describes SCRM as a "multidisciplinary social media approach as it involves all departments in the organization. Instead of only contacting people with sales offers, the aim of social CRM is to build strong relationships with Internet users by giving them a positive experience of the organization's brand, products, and

services." At the end, SCRM turns such social media relationships into loyal customers, which is the main goal, with the product/service purchase being only the secondary role.

The huge shift from "older" electronic CRM to "newer" SCRM and the "nascense of customer management" was illustrated by Harrigan [9]. The customer has become not only an object for measurement and assessment, but also the key player in the relationship, with every activity and behavior monitored online and in relation to their role in the community. Moreover, Osakwe et al. [10] in their study implied that online retail brands should offer their products via social networking sites and also endeavor to keep tracking these online users in the social media community to increase affinity towards their brands.

Recent studies [9,11,12] have pointed out that the nature of SCRM is often affected by its organizational characteristics, mainly size and market orientation. Although SCRM is already considered as a very effective marketing approach, primarily for customer acquisition, we know only little about its application in the business sector in Europe and even less in Central and Eastern European Countries (CEEC). Research investigating businesses in CEEC [13,14] tends to focus more on the traditional CRM approach than on SCRM. A lack of academic research on social media marketing was proved by Klepek´s study [14] which documents that only two papers by Czech researchers in the field of the application of social media for small and medium enterprises were produced between the years 2006 and 2016. Incidentally, one of these two papers was made by authors, so this study follows on it.

Therefore, this study aims to investigate the current level of social customer relationship management in the CEEC region, specifically in the organizations headquartered in the Czech Republic. It deals with issues of SCRM as online communities and its building, management, and use; social media and its use in CRM systems; and customer information and its collection, integration, and use. Moreover, the study also deals with the influence of organizational characteristics—size and market orientation—as factors affecting the level of SCRM.

Harrigan and Morgan [9] argue that small and medium enterprises (SMEs) tend to build their own online communities for customer communication but are less likely to participate in customer-owned communities or to make up such communities for their use in marketing. However, the use of these communities in larger corporations has not yet been researched to a deeper extent. Therefore, the first research question was determined as follows: *What is the level of building of own online communities and participation in relevant customer-owned communities?* Subquestions related to organizational characteristics were formed for this question: *How much is this level affected by organizational size/market orientation?*

The fact that SMEs do not sufficiently participate in communities created by customers goes somehow against the idea designed by Kumar of involving customers [15] but can be explained by lack of time on the side of SMEs. In general, SMEs tend to maintain a higher level of face-to-face contact than larger corporations [16,17]. Moreover, the previous research underlined problems SMEs face in keeping strategic and long-ranged focus [17,18]. Therefore, it can be assumed that the proactive management of interactions in communities and the strategic approach to managing online communities differ according to organization size. Therefore, the second question was phrased as follows: *What is the level of proactive management of interactions in online communities and strategic approach to online communities?* Subquestions related to organizational characteristics were formed for this question: *How much is this level affected by organizational size/market orientation?*

The findings of relevant studies show that social media are an important part of CRM business processes in firms, allowing them have interactions via marketing messages, decisions on products, and conversation in general [19,20]. At the same time, it can be assumed that social media will play a role especially towards final consumers, as relationships with business partners have a rather more long-ranged and personalized character than online interactions in communities. Therefore, the third research question was phrased as follows: *What is the level of use of online communities as a way of engaging*

*with customers?* Subquestions related to organizational characteristics were formed for this question: *How much is this level affected by organizational size/market orientation?*

Social media provide easily accessible customer data used for making strategic marketing decisions [19,21,22]. Despite the fact that such data were always a "motor" driving CRM, the social media data are very different in their nature [23,24]. Moreover, such data are not only accessible and created directly by customers, but also real-time, "messy", and difficult to analyze and quantify [12,19]. Customer data collected on social media are mainly used for decisions by small and medium enterprises and not by larger corporations. While SMEs are strong in customer communication, the larger organizations are better at the information aspects of CRM. Regarding market orientation, differences in the use of social media can be assumed because various market orientations require the establishing of different customer relationships. Therefore, the fourth research question was phrased as follows: *How do organizations use social media in their CRM system to support marketing planning and budgeting and also to analyze responses to marketing campaigns?* Subquestions related to organizational characteristics were formed for this question: *How much is this level affected by organizational size/market orientation?*

The acquisition and management of information is vital for every organization conducting CRM. For SMEs, the social media data in terms of number of likes, tweets, comments, or posts are, in general, too messy and are thus left out of the creation of real customer information and the following marketing decisions [12,19]. Conversely, in larger organizations, the constructs for obtaining the information and its integration are made [18,23]. Such data enable the differentiation of every "customer touch", which is also a source of information [21]. Social media like Facebook, Twitter, and LinkedIn can constitute a number of touch points. In general, the SMEs do not attain more advanced levels in CRM data mining. Carson [25] explains this by the relatively small base of customers and huge number of day-to-day decisions. Studies by Harrigan, Ramsey, and Ibbotson and by Kumar et al. [11,15] found that SMEs do not tend to use customer information for more complex calculations such as lifetime customer value or the value of a customer´s referrals to other customers. Therefore, the fifth research question was phrased as follows: *How do organizations collect, integrate, and use customer information?* Subquestions related to organizational characteristics were formed for this question: *How much is this level affected by organizational size/market orientation?*

## 2. Materials and Methods

The methodology of the paper follows those of previous studies [11,14,21] conducted in the field. The constructed questionnaire, partially tested in previous studies, comprised 54 questions divided into five parts. Part 1 included questions on the importance assigned by the respondents (businesses) to marketing and CRM practices; part 2 included questions on customer relations; part 3 included questions on online communities; part 4 included questions on the use of social networks; part 5 included questions on acquiring and managing information; and, finally, part 6 included questions on the characteristics of businesses (market orientation, size, legal form). This article specifically deals with parts 3, 4, 5, and 6. The majority of the questions were scalable and used a typical five-level Likert scale.

A sample of 362 respondents working in the same number of organizations participated in the study. The data were collected from 248 SMEs with up to 250 employees and 112 firms with over 250 employees during the time period between December 2015 and January 2016. The questionnaire distribution was conducted with the use of 144 reporters (they encouraged organizations to participate and to be responsible in careful completion of the questionnaire by physical visit, phone call, or an e-mail alert). The respondents were offered a summary of the research findings in return for providing their responses, and they were assured that only aggregate results would be reported. The research involved only organizations operating in the Czech Republic. The sample included different sizes and types of businesses; however, the respondents who identified their organization as governmental or nonprofit were removed from the sample.

The sample represents 362 firms out of the 905,706 organizations that—according to the Czech statistical office—actively use social media (4.00%). The majority (58.9%) of examined firms focused on both business and consumer markets. Approximately one-quarter (25.9%) of the firms were oriented solely on business markets, the remainder (11.7%) solely on final consumers.

The statistical significance of results was tested on the 5% level of significance. The statistical software Statgraphics was used for statistical analysis. Firstly, analysis of variance was conducted. The F-test was used to compare standard deviations and the *t*-test was used to compare means. Secondly, in cases where the standardized skewness and/or kurtosis was outside the range of −2 to +2 for one column, indicating non-normality, the Kruskal–Wallis test was conducted and the medians instead of the means were tested.

## 3. Results

### 3.1. Building of Online Communities

Online communities exist in the form of own communities (established and operated by an organization) or customer-owned communities. The involvement of sampled organizations in these two types of communities was not differentiated much: the score for participation in own communities reached 2.89 pts., while in customer-owned communities it was 2.94 pts.

However, the differences according to organization size were more significant. The results show that *own communities* are more frequent in larger firms with over 250 employees. In these larger organizations, the mean value reached 3.17 pts. (on the five-point Likert scale) and was significantly higher than the mean of smaller organizations, where the value was only 2.77 pts. Even higher difference was found in a comparison of organizations with up to 50 employees versus organizations with over 500 employees. Statistical testing confirmed significant statistical variance in the results according to firm size ($p$-value = 0.00132). Participation in *communities owned by customers* is again more frequent in larger organizations. However, the organization size does not affect involvement as much as in the previous case. The score for larger organizations with over 250 employees reached 3.11 pts., while in organizations with up to 250 employees, it was 2.87 pts. In almost one-third of the cases (30.1%), respondents answered that they are "not very sure" or that they "neither agreed nor disagreed". In this case, the differences in results were not statistically significant.

### 3.2. Proactive Management of Interactions in Online Communities and Strategic Approaches to Online Communities

Similar to previous organizational activities, *the proactive management of interactions in communities* is rather the domain of larger organizations. The average score in firms with over 250 employees reached 2.61 pts., whereas in smaller organizations it was 2.21 pts. Interestingly, the highest score was found in organizations sized from 250 to 500 employees (32.4% of respondents strongly or slightly agreed), and not in the largest organizations with over 500 employees where about one-fifth (21.3%) of respondents agreed that they conduct proactive management of interactions. The differences in these results were statistically significant ($p$-value = 0.00823).

*Strategic approach to management of online communities* was recorded more frequently in larger organizations. The average score in organizations with over 250 employees reached 3.17 pts., while in smaller organizations it was only 2.77 pts. This difference was proved to be statistically significant ($p$-value = 0.00483). More than half (55.5%) of the largest firms with over 500 employees agreed with this assertion, while the percentage in agreement in other size groups did not exceed one-third of the respondents.

### 3.3. Online Communities as a Way of Engaging with Customers

Relationships with the customer initiated via online communities are built more frequently in organizations oriented toward final consumers: the average score here reached 2.98 pts. A slightly

lower score of 2.76 pts. was found in firms with hybrid orientation (focusing on both markets). The score was significantly lower—with score of 2.38 pts.—in organizations oriented toward the business market. Differences according to organizational size in the use of online communities as a way of forming relationships with customers were not found.

*3.4. Use of the SCRM System to Support Marketing Planning and Budgeting, Analyze Responses to Marketing Campaigns, and to Customize Customer Communication*

Only 140 (38.7%) organizations from the sample had formally set up and used a CRM system. Therefore, as the number of respondents is lower, the results are interpreted directly according to differences on the Likert scale, and not according to the average score achieved on this scale.

In general, the firms do not *use social media in CRM systems to support marketing planning and budgeting*: only about one-quarter of all respondents replied that they use that for this purpose. There are also some differences regarding the market orientation. Support of social networks for marketing planning and budgeting is used mostly in companies with consumer orientation, where 36.4% of respondents agreed with this assertion. Vice versa, organizations concentrated on the business markets use social media in the CRM system for marketing planning and budgeting much less: only 12.5% of organizations agreed. The differences according to market orientation were found to be statistically significant (*p*-value = 0.00036). Detailed results are available in Table 1.

**Table 1.** Use of social customer relationship management (SCRM) to support marketing according to market orientation.

| | Market Orientation | | | Total |
|---|---|---|---|---|
| | **Business to Business** | **Business to Customer** | **Hybrid** | |
| *N* | *40* | *11* | *89* | *140* |
| 1—strongly disagree | 47.5% | 9.1% | 22.5% | 28.6% |
| 2—slightly disagree | 27.5% | 0.0% | 22.5% | 22.1% |
| 3—neither agree nor disagree | 12.5% | 54.5% | 24.7% | 23.6% |
| 4—slightly agree | 12.5% | 27.3% | 22.5% | 20.0% |
| 5—strongly agree | 0.0% | 9.1% | 7.9% | 5.7% |

Similar to the previous case, differences in the use of *SCRM for marketing campaign effectiveness* according to market orientation were found: significant differences were found (*p*-value = 0.00048), with the organizations with consumer focus using social media for this purpose much more than organizations with business focus.

*SCRM as a factor enabling customization of customer communication* is not affected by organization size. In general, no important differences were found among larger and smaller organizations. Even the more detailed analysis of results, for example, in the comparison of groups of organizations with up to 50 employees and those with over 500 employees, did not prove any differences. Unsurprisingly, differences in the use of social media for customized communication related to market orientation exist. Organizations with a focus on the consumer market use social media for customized communication much more than do those with business market orientation. While the score in consumer-oriented organizations reached 3.27 pts., in business-oriented ones, it reached only 2.48 pts.; these differences are statistically significant (*p*-value = 0.01928).

*3.5. Collecting, Integrating, and Using Customer Information*

In general, *regular collection of customer information* is the most frequent activity associated with social CRM, as it received the highest level of agreement (3.53 pts.). Also, the results show obvious differences in firms affected by the organization size. Organizations with more than 250 employees reached an average score of almost 4 points (3.95 pts.), while smaller ones reached only a score of 3.43 pts. These differences showed strong statistical significance (*p*-value = 0.00001).

*Integration of internal and external customer information* is also relatively very frequent in the examined organizations as the agreement score reached 3.19 pts. In larger firms with over 250 employees, a score of 3.63 pts. was reported, in comparison to smaller ones with an average value of 3.00 pts. The differences between these larger and smaller firms were found to be statistically very significant (*p*-value = 0.00001). At higher or lower organizational size, this difference further increases. More than half of the firms with over 500 employees were identified with such integration. Vice versa, in firms with up to 50 employees, the agreement percentage lower than one-third.

*Use of customer information to assess the lifetime value of customers* is probably not such a frequently applied technique in the sampled organizations as the agreement score reached only 2.95 pts. Lifetime value is calculated more frequently in organizations with over 250 employees, where the average score is 3.40 pts. On the contrary, firms with up to 250 employees reached a much lower average score of 2.75 pts. Again, the differences showed strong statistical significance (*p*-value = 0.000003).

*Customer information can be used to measure the value of each customer´s referrals to other customers.* Differences in the use of customer information to measure the value of customer´s referrals were found; the differences were not so strong as in previous cases but were still statistically significant (*p*-value = 0.02105). Larger firms with over 250 employees reached a score of 3.33 pts., while smaller firms reported an average score of 3.01 pts. Generally, we can say that firms probably do not have much experience with the calculation, as about one-third of them neither agree nor disagree with its use.

## 4. Discussion

The results in this paper offer several insights into the examined SCRM practice to discuss. The results showed that the highest agreement scores were recorded for regular collection of customer information (3.53 pts.) and integration of internal and external customer information (3.20 pts.). On the contrary, the lowest agreement scores were found for the use of online communities as a way of engaging with customers (2.68 pts.) and the use of the SCRM system to support marketing planning and budgeting (2.52 pts.). The remaining values ranged from 2.89 pts. to 3.11 pts.

In regard to organization size, higher agreement scores were recorded for larger organizations (mainly enterprises with more than 500 employees). Also, the larger organizations would rather build their own online communities than participate in customer-owned ones. On the other hand, the use of online communities as a way to make contact with a customer is more frequent in smaller organizations. Interestingly, these findings are in contrast to those of the study by Trainor et al. [26], where no significant association between SCRM capabilities and organization size was found. Advanced calculations such as measuring a customer's life-time value or measuring the value of each customer´s referrals to other customers is rather the domain of larger companies; the smaller ones use them only rarely. In this case, the findings of the previous research by Harrigan, Ramsey, and Ibbotson and by Kumar et al. [11,15] were confirmed. The overall results in relation to organizational size are illustrated in Figure 1.

In regard to market orientation, higher agreement scores were recorded in organizations oriented toward final consumers. The highest differences according to market were found mainly in the use of the SCRM system to support marketing planning and budgeting, where the score in organizations with consumer focus reached 3.27 pts., while organizations with business market orientation reached an average agreement score of only 1.90 pts. Similarly, differences according to orientation were also found in other uses of the SCRM system (customization of customer communication and analyzing responses to marketing campaigns). These findings are in accordance with the results of Cawsey and Rowley´s study [27] which underlines the rather qualitative nature of CRM in business-to-business communication as opposed to the quantitative approach and employment of a number of metrics.

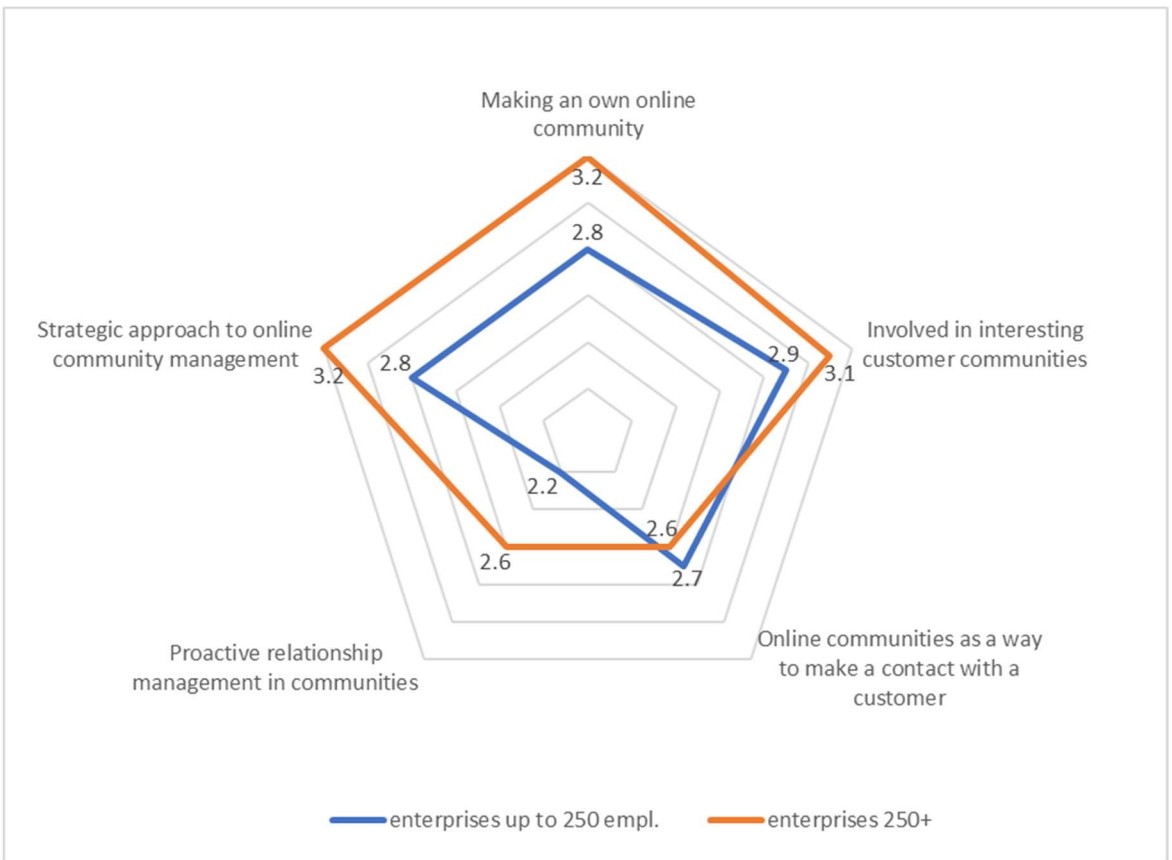

**Figure 1.** Differences in online community management according to organization size.

## 5. Managerial Implications and Conclusions

Several managerial implications can be identified with regards to the present study. Firstly, despite the fact that organization representatives strongly declared that they collect, integrate, and use customer information, they do not use the social media information so often and so fully as possible (this is highly visible especially in the use of social media in CRM systems for which the lowest agreement score was recorded). Secondly, the smaller organizations surprisingly use more online communities as a way to make a contact with a customer. It is likely that in this case, the potential of larger organizations is not fully exploited. Thirdly and finally, market orientation is vital to the level of use of SCRM practices. In this area, SCRM is declared to be useful mainly for organizations with final customer orientation and with only little significance for organizations with business focus.

This study has brought new findings on the SCRM practices of organizations headquartered in the Czech Republic in the following five domains: the building of online communities; the proactive management of interactions in online communities; online communities as a method of customer engagement; use of social media in CRM systems; and the collection, integration, and use of customer information. The paper findings show that SCRM practices are more frequently employed with increasing organization size and also with orientation toward the final consumer. Building upon other research studies [11,15,28], this paper confirmed the significant influence of organizational characteristics such as size and market orientation on SCRM use and can serve as a good basis for the future research conducted for either geographical comparisons or comparisons over time of the state of the art in this field. Future research might be concentrated on other factors affecting the use of SCRM, such as the number of marketing staff, business nature, outsourcing of SCRM activities, and other research interests.

**Author Contributions:** Both authors made equal contributions to the article.

**Funding:** This research received no external funding.

**Acknowledgments:** We acknowledge for administrative and technical support of our home universities as well as to Prof. Viera Pacakova for her technical support, and Dr. Martin Klepek for provision of additional data from his review on social media use.

**Conflicts of Interest:** The authors declare no conflict of interest.

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
