# Peer review of "Social Customer Relationship Management and Organizational Characteristics"

_information, doi:10.3390/info9120306_

Reviewer 1 Report

Dear Authors,

The following aspects should receive your attention:

The aim and objective of the study are missing. Please mention them.

Each of your research questions are, in fact, two questions. Please correct.

The section Discussion and conclusion lacks consistency. Results should be discussed more broadly in the market context.

Conclusions related to the study itself are practically missing.

Author Response

Dear reviewer,

I am reacting point by point (all the changes are highlighted by yellow colour in the text): 

1. The aim and objective of the study are missing. Please mention them.
    One paragraph in between lines 64-69 includes the aim and objective of the study

2. Each of your research questions are, in fact, two questions. Please correct.
    For each research question were formed two subquestions associated with organizational characteristics.

3. The section Discussion and conclusion lacks consistency. Results should be discussed more broadly in the market context.
    This section was completely revised. I hope it provides more consistency right now. Results are discussed also in the market context as it is recommended.

4. Conclusions related to the study itself are practically missing.
Conclusions are available in lines 295-305.

Besides of mentioned, seven references were add to increase the quality of the paper.

Reviewer 2 Report

Authors have made significant improvements to the original submission based on the first review. This includes Introduction, Materials and Methods, Results and Managerial implications and conclusion.

In fact, since the original submission the paper has addressed partly the comments of the reviewer’s. On the basis of these observations this paper provides sufficient contribution to accept for publication after minor correction.

The authors have revised their paper to partly address all my comments.

I am not satisfied with their revision.

The literature review of the manuscript should be even more sufficiently supported by proper references. For this purpose please see following references:
E. Erdei, J. Popp, J. Oláh (2018): Comparison of time-oriented methods to check manufacturing activities and an examination of their efficiency. LogForum, 14(3), 371-386.
J. Oláh, Z. Zéman, I. Balogh, J. Popp (2018): Future challenges and areas of development for supply chain management. LogForum, 14 (1), 127-138.

Author Response

Dear reviewer,

I am reacting on your review,

1. The literature review of the manuscript should be even more sufficiently supported by proper references.

Seven new references were added into the text to increase the quality of the paper (the changes are highlighted by yellow color).

We have also red two literature referrences recommended, however, we have not found a direct link between the papers. Therefore, we did not include them into the paper.

Round  2

Reviewer 1 Report

Dear Authors,

The manuscript has improved compared to previous version.